# Electronic control of redox reactions inside *Escherichia coli* using a genetic module

**Moshe Baruch[1], Sara Tejedor-Sanz[1,2], Lin Su[1,2], Caroline M. Ajo-Franklin[1,2,3]***

**1** The Molecular Foundry, Biological Nanostructures Facility, Lawrence Berkeley National Laboratory, Berkeley, California, United States of America, **2** Department of BioSciences, Rice University, Houston, Texas, United States of America, **3** Institute for Biosciences and Bioengineering, Rice University, Houston, Texas, United States of America

* cajo-franklin@rice.edu

## Abstract

Microorganisms regulate the redox state of different biomolecules to precisely control biological processes. These processes can be modulated by electrochemically coupling intracellular biomolecules to an external electrode, but current approaches afford only limited control and specificity. Here we describe specific electrochemical control of the reduction of intracellular biomolecules in *Escherichia coli* through introduction of a heterologous electron transfer pathway. *E. coli* expressing *cymAmtrCAB* from *Shewanella oneidensis* MR-1 consumed electrons directly from a cathode when fumarate or nitrate, both intracellular electron acceptors, were present. The fumarate-triggered current consumption occurred only when fumarate reductase was present, indicating all the electrons passed through this enzyme. Moreover, CymAMtrCAB-expressing *E. coli* used current to stoichiometrically reduce nitrate. Thus, our work introduces a modular genetic tool to reduce a specific intracellular redox molecule with an electrode, opening the possibility of electronically controlling biological processes such as biosynthesis and growth in any microorganism.

## Introduction

Microorganisms accomplish important biological functions such as conserving energy, regulating gene expression, and powering biosynthesis using different redox-active biomolecules. To enable control of these processes in any microorganism, researchers have coupled the redox state of these biomolecules to an external electrode using membrane-permeable, small molecule redox mediators [1–5], redox polymers [6] and membrane-intercalated nanostructures [7, 8]. These approaches can allow cells to produce electrical current or consume it, resulting in either oxidation or reduction of intracellular redox species, respectively. Bioelectrochemical devices can then be used to drive biosynthetic reactions [1, 4, 5], perform bioelectronic sensing [9], actuate gene expression [3], and modulate cellular growth [10, 11] within the microorganism of interest. Despite these accomplishments, these strategies couple the redox state of the electrode to multiple intracellular redox biomolecules, resulting in off-target effects, cellular toxicity, and poor control of biosynthesis [1, 3, 4]. To achieve precise

**Data Availability Statement:** All relevant data are within the manuscript and its Supporting Information files.

**Funding:** Work at the Molecular Foundry was supported by the Office of Science, Office of Basic

Energy Sciences, of the U.S. Department of Energy under Contract No. DE-AC02-05CH11231. This work was supported by the Office of Naval Research, Award number N000141310551.

**Competing interests:** The authors have declared that no competing interests exist.

electrochemical control of a biological process, a strategy that couples an electrode to a specific intracellular redox pool is still needed [4, 12, 13, 39].

To couple an electrode to specific redox molecules in a bacterium of our choosing, we and others have introduced genes from the Mtr pathway from *Shewanella oneidensis* MR-1 into heterologous bacterial hosts [14–18, 39]. Under anaerobic conditions, *S. oneidensis* can use the Mtr pathway to transfer electrons from catabolism to produce a current at an extracellular electrode (**Fig 1A**). Catabolism of lactate to pyruvate generates electrons that are transferred to the menaquinone (MK) pool either directly or from NADH and Complex I [19]. From MK, electrons traverse the cell envelope through a series of multiheme cyts *c* [20]: CymA in the inner membrane [21], FccA and Stc (also known as CctA) in the periplasm [22], across the outer membrane via the MtrCAB complex [23], and directly from MtrC to an anode [24], typically biased at +200 mV$_{Ag/AgCl}$. Interestingly, the Mtr complex also permits current consumption from a cathode biased at -560 mV$_{Ag/AgCl}$ to drive intracellular fumarate reduction [25]. In this case, electrons are transported to MtrC, across the outer membrane by the MtrCAB complex, to CymA, and finally to the periplasmic fumarate reductase, FccA [25] ultimately reducing fumarate (**Fig 1B**). More broadly, this current consumption opens the possibility to use electricity to reduce $CO_2$ and $N_2$ to fuels and ammonia, respectively, an area of very active research [26, 27].

We have previously shown that oxidation of intracellular lactate can be coupled to current production in an *Escherichia coli* B-strain that heterologously expresses *cymAmtrCAB* [16, 18]. This led us to hypothesize that the Mtr pathway could couple oxidation of a cathode to reduction of intracellular biomolecules. Indeed, other studies [4, 28] have shown cathodic-driven

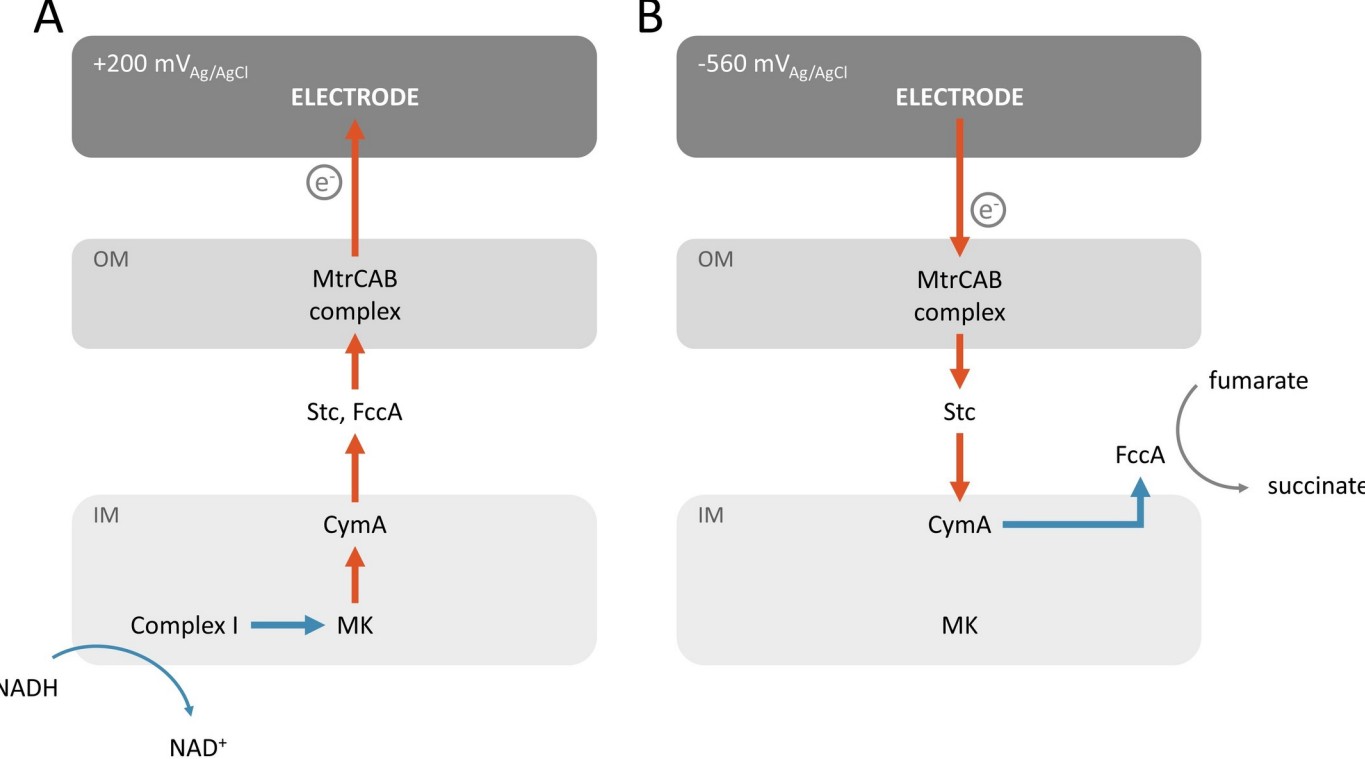

**Fig 1. Coupling of intracellular redox reactions to an electrode in *Shewanella oneidensis* MR-1.** Schematic illustrating the role of the MtrCAB complex and the inner membrane cyt *c* CymA and menaquinone (MK) in the coupling of current production to intracellular oxidation of NADH (A) and current consumption (B) to intracellular reduction of fumarate in *S.oneidensis* MR-1. (OM: outer membrane, IM: Inner membrane).

reduction of intracellular biomolecules in K-strains of *E. coli* expressing portions of the Mtr pathway. However, K-strains and B-strains have significant differences in the cell envelope permeability [29] and type II secretion [30]. Both of these differences will affect electron transfer across the outer membrane, since the cell envelope permeability modulates transit of redox-active biomolecules to the extracellular space [31, 32] and type II secretion is required for cytochromes to be localized to the extracellular leaflet of the outer membrane [30]. Here we probe whether an Mtr-expressing B-strain of *E. coli*, *E. coli* C43(DE3), can directly consume current from a cathode, compare the route of electron flow under anodic and cathodic conditions, and show for the first time that an electrode can stoichiometrically drive the reduction of a specific molecule inside engineered *E. coli*.

## Materials and methods

Methods for S1–S5 Figs can be found in the Supporting information.

### Growth conditions and media composition

All strains, unless otherwise specified, were grown in 2xYT medium at 30°C with 50 μg mL$^{-1}$ kanamycin and 30 μg mL$^{-1}$ chloramphenicol. Strains containing the pAF-*frdABCD* and pAF-*menC* plasmids were also grown with an additional 30 μg mL$^{-1}$ streptomycin. Glycerol stocks were used to inoculate 5 mL 2xYT medium, and cultures were grown overnight at 37°C with 250-rpm shaking. Then, 500 μL of overnight cultures were back-diluted into 50 mL 2xYT medium and grown in a 250 mL flask with 250-rpm shaking for 16 h at 30°C. When the cells reached an OD$_{600}$ = 0.5, 10 μM IPTG was added to induce production of the Mtr pathway, and the cultures were grown at 37°C with 225 rpm shaking overnight.

The M9 media (BD) consists of 6.78 g/L disodium phosphate (anhydrous), 3 g/L KH$_2$PO$_4$, 0.5 g/L NaCl, 1 g/L NH$_4$Cl, and 10 mL/L each of vitamin, amino acid, and trace mineral 100x stock solutions. The 100x vitamin stock solution contained: 2 mg/L D-biotin (B7), 2 mg/L folic acid (B9), 10 mg/L pyridoxine HCl (B6), 5 mg/L thiamine HCl (B1), 5 mg/L nicotinic acid (B3), 5 mg/mL D-pantothenic acid, hexacalcium salt (B5), 0.1 mg/L cobalamin (B12), 5 mg/L p- aminobenzoic acid (PABA), and 5 mg/L α-lipoic acid. The 100x amino acid stock solution (pH 7.0) contained: 2 g/L L-glutamic acid, 2 g/L L-arginine, and 2 g/L D,L-serine. The 100x trace mineral stock solution (pH 7.0) contained: 7.85 mM C$_6$H$_9$NO$_3$Na$_3$, 12.17 mM MgSO$_4$·7H$_2$O, 2.96 mM MnSO$_4$·H$_2$O, 17.11 mM NaCl, 0.36 mM FeSO$_4$·7H$_2$O, 0.68 mM CaCl$_2$·2H$_2$O, 0.42 mM CoCl$_2$·6H$_2$O, 0.95 mM ZnCl$_2$, 0.040 mM CuSO$_4$·5H$_2$O, 0.021 mM AlK (SO$_4$)$_2$·12H$_2$O, 0.016 mM H$_3$BO$_3$, 0.010 mM Na$_2$MoO$_4$·2H$_2$O, 0.010 mM NiCl$_2$·6H$_2$O, and 0.076 mM Na$_2$WO$_4$·2H$_2$O.

The M9 media without ammonia was made from the same materials as the standard M9 medium, except NH$_4$Cl was omitted and the final concentration of NaCl final was increased to 1.5 g/L.

### Plasmids and strains

The strains, plasmids, primers and double stranded DNA fragments used in this study are listed in S1–S4 Tables, respectively. All strains were constructed using *Escherichia coli* strain C43(DE3) (Lucigen, Madison, WI). The deletion of *frdABCD*, *sdhABCD*, *nuoH*, *menA* and *menC* from the C43(DE3) genome was achieved using CRISPR/Cas9 (similar to Pyne et al., [33]) or λ-red recombination [34]. The pEC86 [35] and I5049 [17] plasmids carrying the *E. coli ccm* and *S. oneidensis cymAmtrCAB*, respectively, have been described previously. The pAF-*frdABCD*, pAF-*menC*, and I5105 plasmids were constructed for this work using standard

molecular cloning strategies. *E. coli* strains were grown and prepared for inoculation into bioelectrochemical reactors using standard methods.

## Construction of plasmids

The plasmids used for construction of mutants are presented in S2 Table, and the primers used are listed in S3 Table.

To construct the pAF-*frdABCD* plasmid, we used Gibson assembly to insert the *frdABCD* operon into the pAF001 plasmid. First, we amplified the *frdABCD* operon using the primers "frdABCD+RBS (GB) fw" and "frdABCD (GB) rev" (S3 Table) and using C43(DE3) genomic DNA as a template. The frdABCD+RBS (GB) fw contains both 25 bp of sequence homologous to the pAF001 plasmid and a RBS site. The pAF001 plasmid, which contains the CloDF13 ori, spectinomycin resistance cassette, and a propionate-inducible promoter, was digested with BsaI. The digested plasmid and *frdABCD*-containing fragment were assembled together using Gibson Master Mix (New England BioLabs) as recommended by the manufacturer. The pAF-*frdABCD* plasmid was verified via sequencing.

To construct the pAF-*menC* plasmid, Gibson assembly was also used. The *menC* gene sequence was PCR amplified using primers "menC-fw" and "menC-rev" and C43(DE3) genomic DNA as the template. Both the PCR amplified gene and pAF-*frdABCD* plasmid backbone have been gel purified and then assembled via Gibson Assembly Master Mix as recommended by the manufacturer. The resulting plasmid (pAF-menC) has been PCR amplified using the primers "pAF-menC-fw" and "pAF-menC-rev" (S3 Table) and verified via sequencing.

To construct the I5105 plasmid, we used a gBlock (IDT) composed of the epcD promoter (based on the sequence presented on Boyarskiy et al. 2016 [36]), RBS and the *cymA* sequence flanked by SgrAI and EcoRI digestion sites (S2 Table). The gBlock was then digested by SgrAI and EcoRI. SgrAI and EcoRI were also used to digest I5049 to generate a fragment containing the plasmid backbone and *mtrCAB*. This digested fragment and the digested gBlock were ligated together and transformed into DH5α competent cells (NEB) by electroporation. The resulting plasmid, I5105, containing *cymAmtrCAB* regulated by the ecpD promoter, was verified via sequencing.

## Construction of deletion strains

**CymAMtr-Δfrd.**   The *frd* operon in C43(DE3) *E. coli* was deleted using the CRISPR-Cas9 system. We designed and obtained a gBlock (IDT) composed of a short homologous sequence of *frdA* (spacer) flanked by crispr repeats and a short homologous sequence of pMCC plasmid. The gblock was amplified using PCR and the primers "pMCC(smaI)-crisper fwd" and "pMCC(smaI)-crisper rev." The amplified gBlock was then inserted into the SmaI-digested pMCC plasmid using Gibson assembly and transformed into DH5α competent cells (New England BioLabs) by electroporation. Transformants resistant to chloramphenicol were selected. The plasmid was purified from a selected clone and was sequence verified. Next, a gBlock containing homologous DNA fragments flanking the targeted editing region was designed and obtained. This gBlock was PCR amplified with the primers "Crispr frdA-for" and "Crispr frdD-rev." Then, the pKd46-Cas9 plasmid was introduced into C43(DE3) competent cells by electroporation and transformants resistant to ampicillin were selected. The resulting C43+-pKd46-Cas9 strain was grown in LB supplemented with 10 mM arabinose until OD$_{600}$ = 0.3–0.5 was reached. The cells were then prepared for electroporation and transformed with the pMCC plasmid and the amplified homologous DNA fragments and transformants resistant to ampicillin and chloramphenicol were selected. The knockout of the *frdABCD* operon in the selected clone has been verified via sequencing. These selected clones were grown on agar

plates at 43˚C. The last step was repeated twice, and only clones that lost resistance to chloramphenicol and ampicillin have been selected. Finally, the pEC086 and I5049 plasmids were introduced to the Δ*frdABCD* C43(DE3) background via electroporation and selection for colonies that grow on LB-agar with kanamycin and chloramphenicol.

**CymAMtr-Δ*frd*Δ*sdh*, CymAMtr[s]-Δ*frd*Δ*sdh* and CymAMtr[s]-*frd*[+]Δ*sdh*.** This mutant was constructed using the lambda red mediated gene replacement, adapted from Datsenko and Wanner. The pKd46 plasmid was introduced into Mtr-Δ*frd* mutant by electroporation and transformants resistant to ampicillin were selected. The Mtr-Δ*frd*+pKd46 cells were prepared for electroporation by growing them at 30˚C and adding 10mM L-arabinose when the culture reached $OD_{600} = 0.1$. A DNA linear fragment containing a short 20 bp sequence homologous to the start and end of the *sdhABCD* operon and the pKd4's kanamycin resistance gene flanked by FRT sites was PCR amplified with the primers "sdh-pKd4 fw" and "sdh-pKD3 rev" (S3 Table) using pKD4 as a template. The competent cells were electroporated in the presence of linear fragment and grown on a kanamycin plate at 37˚C. Colonies containing the desired deletion of the *sdhABCD* operon were selected and verified using PCR amplification using the sdh-for and sdh-rev primers (S3 Table). pCP20 was introduced into selected clones by electroporation and transformants resistant to chloramphenicol were selected. These clones were then grown again on agar plates at 43˚C. The last step was repeated twice and a clone that lost resistance to kanamycin, chloramphenicol and ampicillin was selected. The relevant plasmids were introduced to the mutant via electroporation and selection for colonies that grow on LB-agar with antibiotics. The pEC086 and I5049 plasmids were used for the CymAMtr-Δ*frd*Δ*sdh* strain. The pEC086 and I5105 plasmids were used for the CymAMtr[s]-Δ*frd*Δ*sdh* strain. The pEC086, I5105 and pAF-*frdABCD* plasmids were used for the CymAMtr[s]-*frd*[+]Δ*sdh* strain.

**CymAMtr-Δ*nuoH* and Ccm-Δ*nuoH*.** This mutant has been constructed using the lambda red mediated gene replacement. pKd46 plasmid was introduced into C43 (DE3) strain by electroporation and transformants resistant to ampicillin. The C43+pKd46 cells were prepared for electroporation by growing them at 30˚C and adding 10 mM L-arabinose when the culture has reached $OD_{600} = 0.1$. DNA linear fragment containing a short 20 bp sequence of the end and start of the *nuoH* gene and the pKd4's kanamycin resistance gene flanked by FRT sites was PCR amplified with The primers "nuoH-pKd3 fw" and "nuoH-pKd3 rev" (S3 Table) using pKD3 as a template. The competent cells were electroporated with the linear fragment and grown on a kanamycin plate at 37˚C. Colonies containing the desired deletion of the *nuoH* gene have been selected and verified using PCR amplification using the "nuoH-for" and "nuoH-rev" primers (S3 Table). pCP20 was introduced into selected clones by electroporation and transformants resistant to chloramphenicol were selected. Cloned has been selected and grown again on agar plates at 43˚C. The last step has been repeated twice and a clone that lost resistance to kanamycin, chloramphenicol and ampicillin has been selected. The relevant plasmids were introduced to the mutant via electroporation and selection for colonies that grow on LB-agar with antibiotics. The pEC086 and I5105 plasmids were used for the CymAMtr-Δ*nuoH* strain. The pEC086 and I5023 plasmids were used for the Ccm-Δ*nuoH* strain.

**CymAMtr-Δ*menA*.** This mutant has been constructed using the lambda red mediated gene replacement. pKd46 plasmid was introduced into C43 (DE3) strain by electroporation and transformants resistant to ampicillin. The C43+pKd46 cells were prepared for electroporation by growing them at 30˚C and adding 10 mM L-arabinose when the culture has reached $OD_{600} = 0.1$. DNA linear fragment containing a short 20 bp sequence of the end and start of the *menA* gene and the pKd4's kanamycin resistance gene flanked by FRT sites was PCR amplified with the primers "menA-pKd4-fw" and "menA-pKd4- rev" (S3 Table) using pKD3 as a template. pCP20 was introduced into C43 (DE3) strain by electroporation and transformants resistant to kanamycin. Colonies containing pCP20 have been growing at 30˚C and

screened for loss of resistance to kanamycin on plates at 37˚C. The selected colonies were further grown on plate at 43˚C and screened for loss of resistance to chloramphenicol. The resulting strains were tested using PCR using the primers "menA-rev" and "menA-fw" (S3 Table). The pEC086 and I5049 plasmids were introduced to the mutant via electroporation and selection for colonies that grow on LB-agar with antibiotics.

**CymAMtr-*ΔmenC* and CymAMtr-+*menC*.**   This mutant has been constructed using the lambda red mediated gene replacement. pKd46 plasmid was introduced into C43 (DE3) strain by electroporation and transformants resistant to ampicillin. The C43+pKd46 cells were prepared for electroporation by growing them at 30˚C and adding 10 mM L-arabinose when the culture has reached $OD_{600}$ = 0.1. DNA linear fragment containing a short 20 bp sequence of the end and start of the *menC* gene and the pKd4's kanamycin resistance gene flanked by FRT sites was PCR amplified with the primers "menC-pKd4-fw" and "menC-pKd4- rev" (S3 Table) using pKD3 as a template. pCP20 was introduced into C43 (DE3) strain by electroporation and transformants resistant to kanamycin. Colonies containing pCP20 have been growing at 30˚C and screened for loss of resistance to kanamycin on plates at 37˚C. The selected colonies were further grown on plates at 42˚C with kanamycin and ampicillin to find sensitive strains to these antibiotics. The resulting strains were tested using PCR using the primers "menC-test-rev" and "menC-test-fw" (S3 Table). The relevant plasmids were introduced to the mutant via electroporation and selection for colonies that grow on LB-agar with antibiotics. The pEC086 and I5049 plasmids were used for the CymAMtr-*ΔmenC* strain. The pEC086, I5049 and pAF-menC plasmids were used for the CymAMtr-+*menC*.

The pEC086 and I5049 plasmids were introduced to the mutant via electroporation and selection for colonies that grow on LB-agar with antibiotics.

## Electrochemical measurements of *E. coli* strains in bioreactors

All electrochemical measurements were performed in potentiostat-controlled (VMP300, Bio-Logic LLC), three-electrode, custom-made bioelectrochemical reactors (Adams & Chittenden, Berkeley, CA) that used a cation exchange membrane (CMI-7000, Membranes International, Ringwood, NJ) to separate two 250 mL chambers. The working electrode was a 25×25 mm piece of graphite felt with a piece of Ti wire threaded vertically and the counter electrode was a piece of Ti wire. For reference we used a 3M Ag/AgCl reference electrode (CHI111, CH Instruments, Austin). The working and the reference electrode were placed in one chamber and the counter electrode was placed in the second chamber. Both chambers were filled with 140 mL M9 media and autoclaved at 121˚C for 20 min. After autoclaving, filter-sterilized solutions of vitamins, minerals, amino acids, 50 $mgL^{-1}$ kanamycin, and 10 μM IPTG were added to the working electrode chamber.

Throughout the experiment, the environmental conditions within the bioreactors were carefully controlled. The bioreactors were incubated at 30˚C. The working electrode chamber was continuously sparged with $N_2$ gas and was stirred using a magnetic stirrer rotating at ~200 rpm. The working electrode was biased to +0.200 $V_{Ag/AgCl}$ and current was monitored using a potentiostat (VMP300, Bio-Logic LLC). After the baseline current stabilized (~4 h), *E. coli* cells in fresh M9 were introduced into the bioreactor to a final cell density of 0.6 $OD_{600}$. After 3 days of incubation (unless otherwise noted), the working electrode potential was switched to -0.560 $V_{Ag/AgCl}$. Fumarate was added to 50 mM and nitrate was added to 10 mM. Pyruvate was present at 40 mM only for the 14-day long experiments shown in Fig 5. Each experiment was replicated across three technical replicates and two biological replicates. Biological replicates are experiments employing a different culture of the same strain. Technical replicates are experiments using the same culture in different bioelectrochemical reactors.

Spent media and cell samples were removed from the bioreactors for subsequent analysis. Spent media was collected from the working electrode chamber using a sterile needle. These samples were centrifuged at 5,000 g for 5 min to pellet any planktonic cells, and the supernatant was analyzed for the presence of small molecules with HPLC (see following section). To extract cells, the bioreactors were depolarized, gently shaken to remove the cells attached to the working electrode, and the resulting suspension was analyzed for cytochrome c content via enhanced chemiluminescence and cell density via $OD_{600}$ and colony forming units (refer to SI for additional details).

## Detection of organic acids and ammonia

From the supernatant samples, the concentration of various organic acids was measured by HPLC (Agilent, 1260 Infinity), using a standard analytical system (Shimadzu, Kyoto, Japan) equipped with an Organic Acid Analysis column (Bio-Rad, HPX-87H ion exclusion column) at 35˚C. The eluent was 5 mm sulfuric acid, used at a flow rate of 0.6 mL min$^{-1}$ and compounds were detected by refractive index. A five-point calibration curve based on peak area was generated and used to calculate concentrations in the unknown samples. For determination of ammonia concentrations, we employed assay kits (Sigma, AA0100) according to the manufacturer's protocols.

## Calculation of succinate production based on the measured cathodic current

In each experiment, the current from three polarized reactors was measured. The total charge in each reactor was integrated and converted to moles of electrons. Based on the volume of the reactor (140 mL) and the that two electrons are needed to reduce fumarate to succinate, these moles of electrons were converted to a change in the succinate concentration. Finally, the succinate concentration was divided by 14 days (the duration of the experiment).

## Calculation of ammonium production based on the measured cathodic current

To predict how much ammonia would accumulate in the reactor, we first integrated the current over the 14 days period to determine the total charge in Coulombs consumed from each cathode, denoted $Q$. To calculate the amount of ammonia produced by cathodic current, we posited that two moles of electrons from the cathode are used by the cytoplasmic NarGHI complex to reduce 1 mole of nitrate to nitrite. There are two scenarios for reduction of nitrite to ammonia. Either the Nir complex could use intracellular NADH to reduce nitrite to ammonia, without additional cathodic electrons. From this scenario, we expect the change in ammonia concentration over 14 days in the 140 mL reactors should be:

$$\Delta[ammonia] = Q \times \frac{1 \; mole \; electrons}{96485 \; Coulombs} \times \frac{1}{0.140 \; L} \times \frac{1 \; mole \; ammonia}{2 \; moles \; electrons} \qquad \text{Eq 1}$$

Alternatively, nitrite could be reduced to ammonia by the Nrf complex and six cathodic electrons, so that a total of eight cathodic electrons are used to produce ammonia from nitrate. In this second scenario, we expect the change in ammonia concentration over 14 days in the 140 mL reactors should be:

$$\Delta[ammonia] = Q \times \frac{1 \; mole \; electrons}{96485 \; Coulombs} \times \frac{1}{0.140 \; L} \times \frac{1 \; mole \; ammonia}{8 \; moles \; electrons} \qquad \text{Eq 2}$$

## Results

### *E. coli* consume current using Mtr and native oxidoreductases

We first sought to determine if the Mtr pathway could allow cathodic electrons to directly enter a heterologous host upon addition of an electron acceptor. Since *E. coli* has two MK-linked fumarate reductases, FrdABCD and SdhABCD, we hypothesized that the Mtr pathway in *E. coli* could deliver cathodic electrons via these native proteins to fumarate (Fig 1B). To probe the specific role of the Mtr pathway, we compared the behavior of several strains: *E. coli* expressing only the cytochrome *c* maturation (*ccm*) genes (abbrev. Ccm-*E.coli*) [17], *E. coli* expressing *ccm* and *mtrCAB* (abbrev. Mtr-*E. coli)* [15], and *E. coli* expressing *ccm* and *cym-AmtrCAB)* (abbrev. CymAMtr-*E. coli)* [17]. The *ccm* genes are required to make cytochromes *c* in the C43(DE3) parental background.

To prepare *E. coli* for cathodic conditions, individual strains were first grown aerobically, incubated in potentiostatic-controlled bioreactors under anaerobic conditions. Then we exposed them to anodic conditions ($\Delta V = +200\ mV_{Ag/AgCl}$) for at least one day to exhaust any possible internal electron storage that could compete with the cathode as electron donor and promote cell attachment to the electrode. This acclimation period also yielded more reproducible results. Under these conditions, the CymAMtr-*E. coli* strain produced a significant steady-state current, while Ccm-*E. coli* and Mtr-*E. coli* produced much lower currents (Fig 2A), reinforcing that CymA is important for current production [16, 18]. As anaerobic conditions were maintained, the electrode bias was then switched to cathodic conditions ($\Delta V = -560\ V_{Ag/AgCl}$), fumarate was added, and current consumption was measured. In the absence of *E. coli*, neither current consumption nor fumarate reduction was observed (S1A Fig). Likewise, the Ccm-*E. coli* strain did not consume significant levels of current (Fig 2B). In contrast, both the Mtr-*E. coli* and the CymAMtr-*E. coli* strains consumed significant levels of current (Fig 2B), starting within 30 seconds after fumarate addition (S1B Fig). This rapid onset, compared with the ~30 m required for gene expression, indicates that a change in gene expression is not required to initiate current consumption. There is no significant difference in the MtrCAB abundance in the CymAMtr-*E. coli* and Mtr-*E. coli* [17], so we can rule out a difference in gene expression as

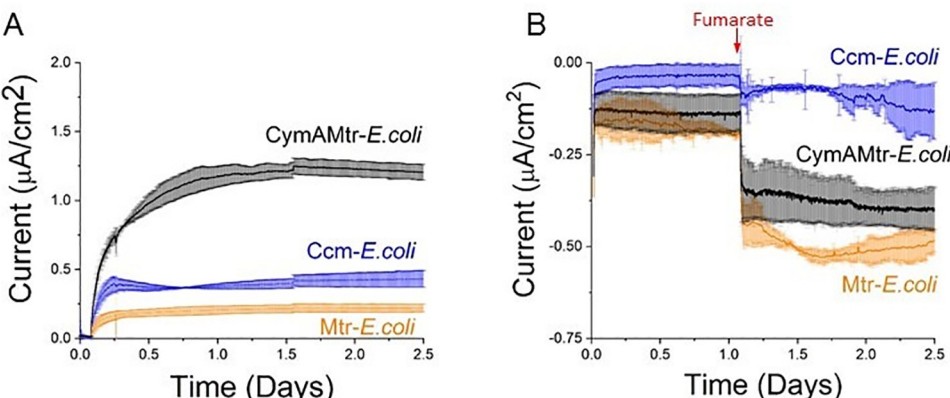

**Fig 2. Expression of *mtrCAB* from *S. oneidensis* MR-1 allows *Escherichia coli* to directly produce or consume current.** (A) Chronoamperometry of bioelectrochemical reactors containing Ccm-*E. coli* (orange), Mtr-*E. coli* (orange), or CymAMtr-*E. coli* (black) in the anodic compartment with the anode poised to +200 mV$_{Ag/AgCl}$. Lactate is provided as an electron donor and the anodic chamber is kept anaerobic by bubbling with N$_2$(g). (B) Chronoamperometry of bioelectrochemical reactors containing Ccm-*E. coli* (orange), Mtr-*E. coli* (orange), or CymAMtr-*E. coli* (black) in the cathodic compartment with the cathode poised to -0.56V$_{Ag/AgCl}$. Addition of 50 mM fumarate is indicated by the red arrow. The error bars indicate the standard deviation in current from three bioreactors.

the origin for the current difference. Thus, CymA is not required for anaerobic current consumption in *E. coli*. This contrasts with its role in *S. oneidensis* where CymA is required for current consumption under anaerobic but not under aerobic conditions [37]. NapC, homologous to the CymA protein of *S. oneidensis*, is disrupted in the C43(DE3) background [38], so it cannot be involved in current consumption. Rather, MtrCAB either directly or more probably, indirectly through as-yet-unknown native biomolecule inside *E. coli*, enables new host microorganisms to directly accept electrons from a cathode. To maintain optimal metabolic activity in the strains during the anodic acclimation, we used CymAMtr-*E. coli* in the rest of our experiments.

Cyclic voltammetry of the CymAMtr-*E. coli* strain with fumarate (S1D Fig) revealed negative shift of the catalytic wave starts at -43 mV vs. Ag/AgCl, which is close to the redox potential of FrdAB [39], suggesting that electrons entering the Mtr pathway could transfer to the *E. coli* fumarate reductase. To determine whether all cathodic electrons passed through the native fumarate reductases of *E. coli* upon fumarate addition, we compared the amount of current consumed by CymAMtr-*E. coli* in the wt, *ΔfrdABCD* (abbrev. CymAMtr-*Δfrd*), and *ΔfrdABCDΔsdhABCD* (abbrev. CymAMtr-*ΔfrdΔsdh*) backgrounds. The CymAMtr-*Δfrd* strain consumed ~40% as much current as the CymAMtr-*E. coli* strain (Fig 3A) and the CymAMtr-*ΔfrdΔsdh* strain did not consume any significant current (Fig 3B). These observations strongly suggest that cathodic-derived electrons pass solely through FrdABCD and SdhABCD upon fumarate addition in *E. coli* expressing *mtrCAB*.

To confirm that the inability of CymAMtr-*ΔfrdΔsdh* to uptake electrons was due only to deletion of *frd* and *sdh*, we probed current consumption in strains with complemented expression of *frdABCD*. To do so, we first altered the regulation of the *cymAmtrCAB* operon to accommodate expression of *frdABCD* from a third plasmid. This created the parental CymAMtr^S^-*ΔfrdΔsdh* strain and the complemented CymAMtr^S^-*frd^+^Δsdh* strain, which could reduce fumarate and express CymA MtrCAB (S2C Fig). The CymAMtr^S^-*frd^+^Δsdh* strain consumed a significant current upon fumarate addition (Fig 3C), in contrast to the CymAMtr^S^-*ΔfrdΔsdh* strain, which did not consume any current. These data show the Mtr pathway delivers cathodic electrons via the MK-linked fumarate reductases to fumarate in *E. coli*. More broadly, cathodic current flows through only FrdABCD in the Mtr^S^-*frd^+^Δsdh* upon addition of fumarate, which is the first demonstration to our knowledge of a heterologous genetic module that directs electrons to only a single native oxidoreductase inside a bacterial strain.

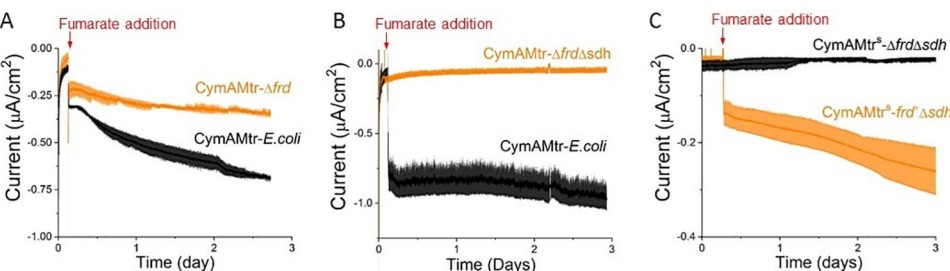

**Fig 3. Fumarate-triggered current consumption in CymAMtr-*E. coli* requires *E. coli* fumarate reductases.** (A) Current consumption by the CymAMtr-*E. coli* in the wt (black) and *ΔfrdABCD* (orange) background upon addition of fumarate, showing reduced current consumption in the *ΔfrdABCD* background. (B) Current consumption by CymAMtr-*E. coli* in the wt (black) and *ΔfrdABCD ΔsdhABCD* (orange) backgrounds upon addition of 50 mM fumarate, showing that current is not consumed when fumarate reductase is absent. (C) Current consumption by CymAMtr-E. coli in the *ΔfrdABCD ΔsdhABCD* (black) and *frdABCD*-complemented *ΔsdhABCD* background (orange). All the experiments were performed under anaerobic conditions with the cathode poised to -560 mV$_{Ag/AgCl}$. Addition of fumarate is indicated by the red arrow, and the bars indicate the standard deviation in current from three bioelectrochemical reactors.

## Menaquinone and Complex I are essential for coupling intracellular oxidations to an anode, but not for coupling reductions to a cathode

Since a MK-linked fumarate reductase is essential for current consumption under cathodic conditions, it is likely that cathodic electrons flow through a quinone. To test this hypothesis, we examined the bioelectrochemical behavior of two strains expressing *cymAmtr* that lack genes essential for menaquinone synthesis, *menA* [40] (abbrev. CymAMtr-Δ*menA)* and *menC (CymAMtr-ΔmenC)* [41] (S4A and S4C Fig). *menC* is also essential for synthesis of the quinone-derived redox shuttle ACNQ [42], allowing us to also test whether ACNQ is an electron carrier here. As before, these strains were acclimated in bioelectrochemical reactors under anodic conditions with similar cell densities before being switched to cathodic conditions.

Under anodic conditions, current production by the CymAMtr-Δ*menC* and CymAMtr-Δ*menA* strains significantly declines to near the current levels produced by the Ccm-*E. coli* strain (Fig 4A and 4C). Complementation of the *menC* in trans (S4B and S4C Fig) restores the current production to the CymAMtr-*E. coli* levels (Fig 4C). These observations demonstrate that menaquinone mediates electron flow from the cytosol to CymA in *E. coli* just as in

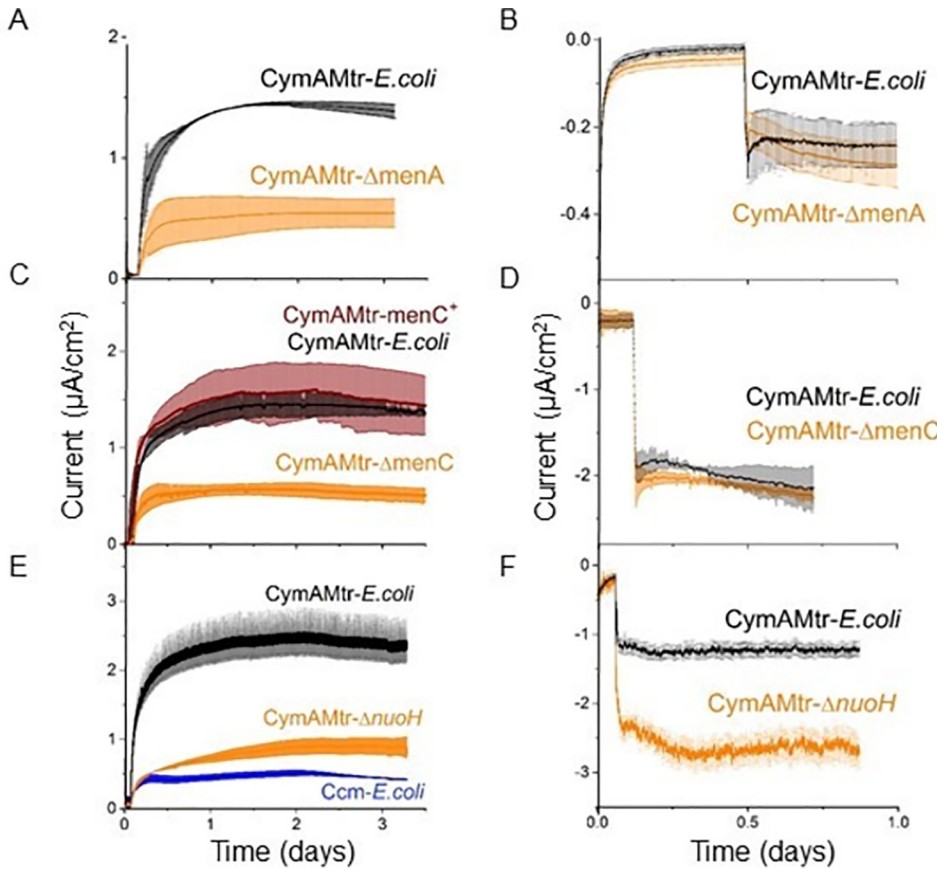

**Fig 4. Mtr-expressing *E. coli* requires menaquinone and Complex I to generate current, but does not require them to consume current.** (A,C,E) Current production under anodic conditions and (B,D,F) current consumption under cathodic conditions for the CymAMtr-*E. coli* in wt (black) and gene deletion (orange) backgrounds. (A,B) Current as a function of time for CymAMtr-*E.coli* in the (A,B) Δ*menA* background and (C,D) Δ*menC* background (orange). Complemented strain *menC*+ is shown in red. (E) Current as a function of time for CymAMtr-*E. coli* in the Δ*nuoH* background. The anode was poised to +200 mV$_{Ag/AgC}$ and the cathode was poised to -560 mV$_{Ag/AgCl}$ for all experiments. Fumarate was added to 50 mM, and the error bars indicate the standard deviation in current from triplicate bioelectrochemical reactors.

*S. oneidensis* MR-1 [43, 44]. Under cathodic conditions, the current consumption by Cym-AMtr-expressing *E. coli* in the Δ*menA* and Δ*menC* strains was not significantly different from the wt background (Fig 4B and 4D). These observations indicate that the current consumption in Mtr-expressing *E. coli* does not rely on the presence of menaquinone or ACNQ, in contrast to *S. oneidensis* [25, 39] and other reports in *E. coli* [28]. In CymAMtr-*E. coli*, free MtrA can be readily isolated from the periplasm without MtrC [17]. Thus, we suggest the fumarate reductase of *E. coli* accepts electrons either directly from free MtrA or indirectly through as-yet-unknown native biomolecule inside *E. coli*. A possible membrane molecule candidate could be ubiquinone, as it has been reported to be a component of the aerobic inward extracellular electron transfer chain in *S. oneidensis* [37].

While fumarate reductase accepts cathodic electrons from the Mtr pathway, we also assume it is still able to accept electrons from $MKH_2$. Even when carbon sources that contribute to the $MKH_2$ are absent from the reactor, CymAMtrA-*E. coli* can produce current for a few days in the absence of an electron donor [16], suggesting this strain stores and slowly utilizes reducing equivalents. Interestingly, we observed that the longer CymAMtr-*E. coli* was deprived of a carbon source in our bioreactors (and thus the lower the stored reducing equivalents), the higher current consumption was upon fumarate addition (S3A Fig). Taken together, these observations suggested that electrons in the MK pool compete with cathodic electrons for fumarate reductase. Since Complex I (NDH-1) catalyzes the transfer of electrons from NADH to $MKH_2$ under anaerobic conditions [45], we probed the effect of disrupting Complex I, a hypothetical source of competing electron donors, on current consumption. We prepared CymAMtr-*E. coli* lacking a functional NDH-1 [46, 47], abbrev. CymAMtr-Δ*nuoH* and acclimated this strain in bioreactors as before.

Under anodic conditions, the CymAMtr-Δ*nuoH* strain produced only ~33% as much current as the CymAMtr-*E. coli* and only slightly more current than the Ccm-*E. coli* (Fig 4E) Under cathodic conditions, the CymAMtr-Δ*nuoH* strain consumed ~225% more current than the CymAMtr-*E. coli* upon fumarate addition (Fig 4F). An analysis of cyt c present in the whole cell lysates of CymAMtr-Δ*nuoH* showed that expression of MtrA and MtrC was significantly lower in the deletion strain (S3B Fig) compared to the CymAMtr-*E. coli*. We suggest that this lower expression is a downstream effect of growth defects observed in Complex I deletion mutants [48]. Nonetheless, this observation rules out that higher levels of Mtr cyt *c* in the Δ*nuoH* background causes the increased current consumption. Instead, these data strongly suggest that eliminating a competing electron flux into fumarate reductase allows additional cathodic electrons to enter the Mtr pathway.

## Current consumption by Mtr-expressing *E. coli* yields non-stoichiometric accumulation of succinate

Having demonstrated that current consumption in *E. coli* requires *mtrCAB* and fumarate reductase, we turned to the question of whether cathodic electrons transported by the Mtr pathway could stoichiometrically drive intracellular reduction of fumarate to succinate. We chose to use the CymAMtr-Δ*nuoH* strain in reactors for subsequent experiments due to its higher electron uptake capacity. We monitored the CymAMtr-Δ*nuoH* strain in reactors poised to cathodic conditions (polarized reactors) and into reactors which were not connected (unpolarized reactors) and measured the extracellular concentrations of several organic acids after addition of fumarate. We found it necessary to supplement the reactors with 40 mM pyruvate, a fermentable carbon source, to sustain bacterial viability during the 14-day long experiment. Pyruvate by itself did not trigger any current consumption (S5A Fig) and did not introduce additional electrode-coupled reactions in bioelectrochemical reactors, indicating that it did

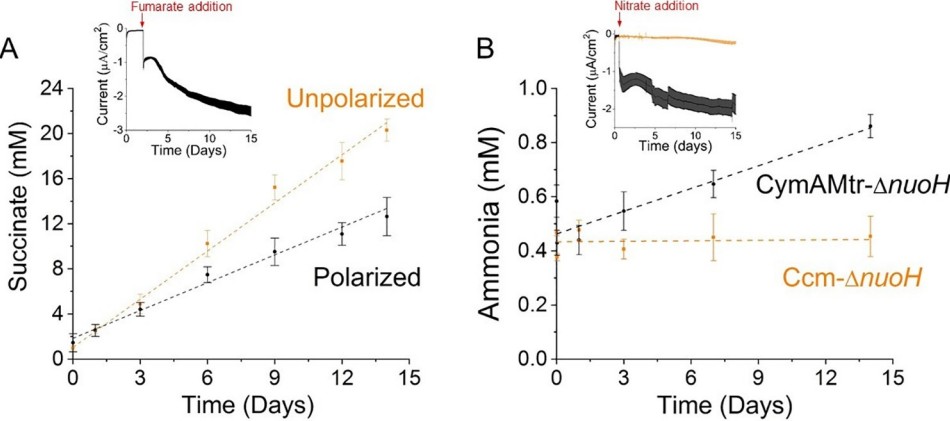

**Fig 5. Current consumption is coupled to non-stoichiometric reduction of fumarate, but stoichiometric reduction of nitrate.** (A) The concentration of succinate in the extracellular media as a function of time in polarized (black) and unpolarized (orange) bioelectrochemical reactors containing the CymAMtr-*ΔnuoH* strain. Dashed lines indicate the linear trends for the polarized (black, $R^2$–0.98) and unpolarized (orange, $R^2$–0.99) reactors, displaying a slope of 0.82 mM succinate day$^{-1}$ and 1.42 mM succinate day$^{-1}$, respectively. (inset) Current consumption in the polarized reactors upon fumarate addition. (B) The concentration of ammonia in the extracellular media as a function of time in bioelectrochemical reactors containing the CymAMtr-*ΔnuoH* strain (black) or Ccm-*ΔnuoH* (orange) after addition of 10 mM nitrate. Dashed lines indicate the linear trend for the ammonia production by CymAMtr-*ΔnuoH* (black, $R^2$–0.88) displaying a slope of 0.027 mM ammonia day$^{-1}$. The ammonia concentration for the Ccm-*ΔnuoH* strain (orange line provided to guide the eye) does not change significantly. In all experiments, the cathode is poised to -560 mV$_{Ag/AgCl}$ and the error bars indicate the standard deviation in current from triplicate bioelectrochemical reactors.

not directly affect our electrochemical measurement. Notably, since lactate dehydrogenase reduces pyruvate to lactate using NADH under anaerobic conditions, the absence of current consumption also indicates that the cathode cannot supply reducing equivalents in place of NADH.

The polarized reactors steadily consumed current and accumulated succinate at a different rate than the unpolarized reactions (Fig 5A). The accumulation of succinate depends on both its production and consumption. Thus, we can estimate the expected accumulation of succinate from the consumed current, if we assume succinate consumption is equal under both polarized and unpolarized conditions. With this assumption, we used the total current consumption to estimate that an additional 0.49 mM succinate would accumulate in the polarized reactors over 14 days compared to unpolarized reactors. However, we observed that the polarized reactors accumulated 29% less succinate than the unpolarized reactors, 12.63 ±1.68 mM vs 20.28 ±0.97 mM over 14 days, respectively. Thus, the number of electrons accumulated in succinate is opposite in direction and 10-fold higher in magnitude than what we expected. Overall, the concentrations of other organic acids we monitored were very similar in the polarized and unpolarized reactors (S5C Fig). Only the formate concentration was slightly higher in the polarized reactor by 2.38 ±0.05 mM after 9 days (S5C Fig), but this minor change is insufficient to explain the dramatic difference between the expected and observed succinate accumulation. Surprisingly, when we repeated this experiment with the CymAMtr-*E. coli* strain we did not detect any significant difference between the polarized and unpolarized reactors (S5D Fig) suggesting that the higher current consumption by the CymAMtr-*ΔnuoH* is essential for this phenotype.

Since we have established that fumarate is only being reduced by fumarate reductases in Mtr-expressing *E. coli* (Fig 2), these observations suggest that the assumption that succinate is consumed equally in the polarized and unpolarized bioelectrochemical reactors is incorrect. Succinate is an intermediate in the TCA cycle, providing many opportunities for its

consumption. Moreover, succinate is a substrate for enzymes that are both allosterically and transcriptionally regulated. Transcriptional regulation is governed by the redox state of the cell, which is likely to be different in polarized and unpolarized conditions. Thus, we speculate that the non-stoichiometric accumulation of succinate results from differences in its consumption under polarized and unpolarized conditions.

## Current consumption by Mtr-expressing *E. coli* yields stoichiometric reduction of nitrate

Our postulate that non-stoichiometric accumulation of succinate was due to unequal consumption led us to examine whether intracellular reductions could be stoichiometrically driven via other oxidoreductases where the product is not utilized by E. coli under our experimental conditions. We chose to focus on nitrate reduction because under our reactor conditions *E. coli* does not grow, thus we expect that ammonia, the product of nitrate reduction, will not be used in assimilation. In the C43(DE3) background, we expect the first step of this reduction, the reduction of nitrate to nitrite, to be cytoplasmic. The periplasmic nitrate reductase Nap is deleted in C43(DE3) [36], leaving only the NarGHI cytoplasmic nitrate reductase that utilizes $MKH_2$ as an electron donor [47]. The second step, the reduction of nitrite to ammonia, can be catalyzed by either Nrf periplasmic complexes or cytoplasmic Nir enzyme complexes [48, 49]. When the availability of nitrate exceeds the ability of *E. coli* to consume it, the Nir enzyme complex is primarily responsible for nitrite reduction and uses NADH as a source of reducing equivalents [50]. Alternatively, when nitrate is present at low amounts, i.e. <1 mM, the Nrf complex is the predominant nitrate reductase and uses $MKH_2$ to produce nitrite [50].

To establish whether nitrate could be reduced by the Mtr pathway in *E. coli*, we added nitrate to reactors without bacteria, with Ccm-Δ*nuoH*, and with CymAMtr-Δ*nuoH* and monitored current flow. No current was consumed in the reactors without bacteria, confirming that nitrate was not abiotically reduced. While low levels of current were consumed upon nitrate (Fig 5B, inset) addition to reactors containing the Ccm-*E. coli*, the CymAMtr-*E. coli* consumed ~5.6 fold higher current levels, respectively, indicating that the majority of the cathodic electron flux in CymAMtr-*E. coli* is Mtr-dependent. Nitrate addition stimulates equal levels of current consumption by CymAMtr-E.coli and CymAMtr-ΔmenC strains (S6 Fig), indicating menaquinone is not required for this electron transfer. These data provide an additional example of delivery of electrons to inner membrane oxidoreductase, NarGHI, in a heterologous host by *mtrCAB and suggest electrons transit a similar route to different MK-linked oxidoreductases.*

To probe whether nitrate could be reduced stoichiometrically using cathodic electrons, we measured ammonia accumulation in reactors containing the Ccm-*E. coli* and CymAMtr-*E. coli* strains over 14 days (Fig 5B). While the ammonia concentration in reactors containing the Ccm-*E. coli* did not change significantly, the reactors containing the CymAMtr-*E. coli* accumulated more ammonia at a steady rate, and over 14 days, accumulated $0.40 \pm 0.09$ mM more ammonia (Fig 5B). These observations indicate that cathodic current through the Mtr complex is linked to reduction of nitrate.

To determine the quantitative relationship between cathodic current and ammonia accumulation, we used the total current consumed to calculate the expected increase in ammonia concentration (see Methods for details). We posited that two cathodic electrons are used instead of $MKH_2$ as an electron donor for NarGHI-catalyzed reduction of nitrate. Since a high concentration of nitrate (10 mM) was added to the reactors, nitrite reduction will be predominately catalyzed by the Nir complex, which utilizes NADH as its native electron donor. Our prior observation that addition of pyruvate does not trigger current consumption strongly

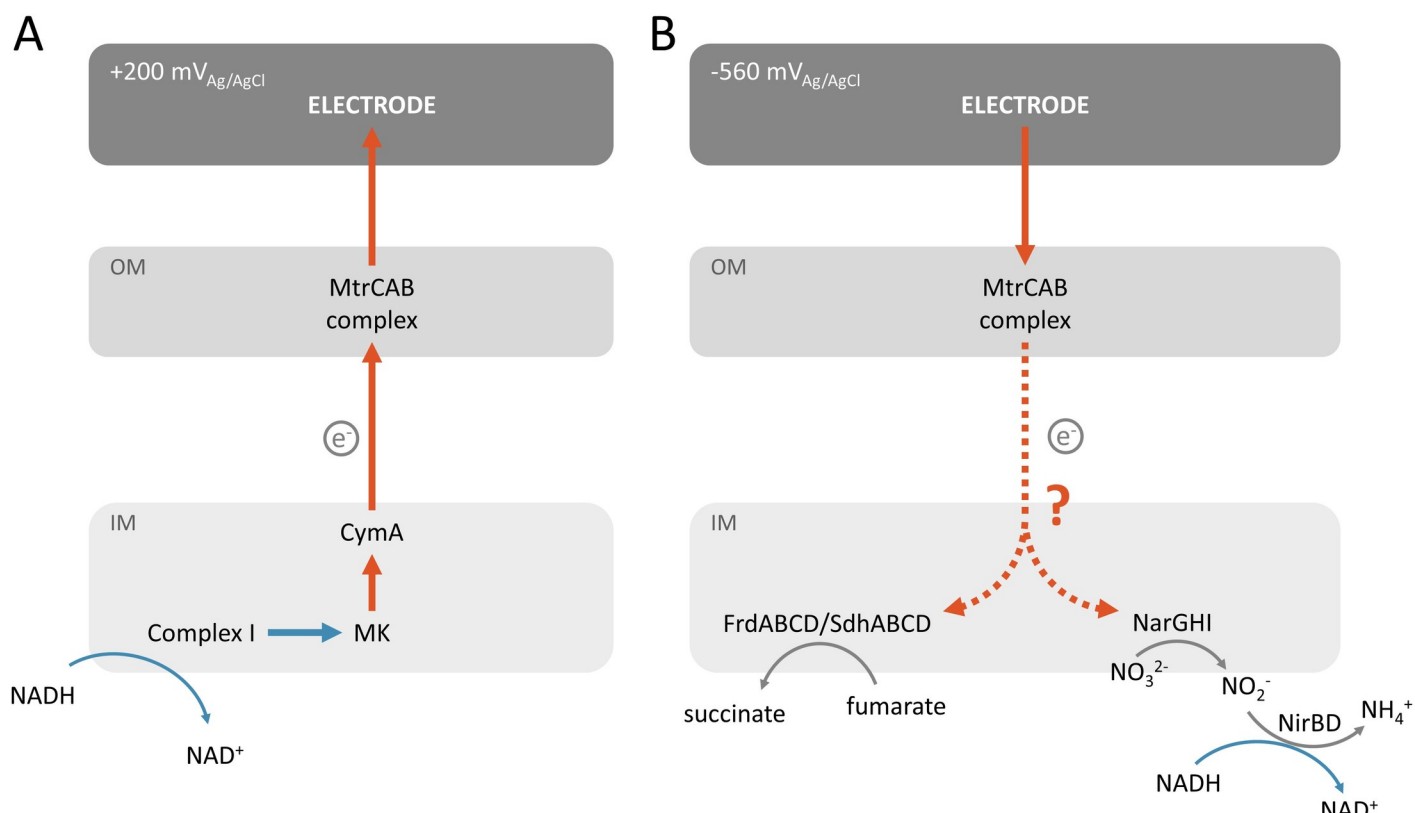

**Fig 6. Model for how heterologously expressed MtrCAB couples intracellular oxidations (left) and reductions (blue arrows) to current production and consumption, respectively, in *E. coli*.** (A) Oxidation of NADH by Complex I transfers electrons to MK. Electrons flow from MK via CymA to the MtrCAB complex. (B) MtrCAB transfer electrons via an unknown process to FrdABCD/SdhABCD and the NarGHI enzymes, which in turn reduce fumarate and nitrate to succinate and nitrite, respectively. Further reduction for the nitrite to ammonia is mediated through NirBD.

suggests that cathodic electrons cannot replace NADH, so only a total of two moles of cathodic electrons will be consumed per mole of ammonia (Eq 1). Using this assumption, we calculate that 0.49 mM ammonia will accumulate over the 14 day period, in excellent agreement with the observed ammonia concentration (0.40 ± 0.09 mM). In contrast, if we assume that cathodic electrons can replace NADH as the electron donor (Eq 2), the predicted ammonia concentration (0.12 mM) is ~one-third the measured concentration. Thus, the excellent agreement between the observed and expected changes in ammonia accumulation indicate that cathodic electrons delivered through the Mtr pathway were used to stoichiometrically reduce nitrate to nitrite. More broadly, these data indicate that *mtrCAB* is a genetic module that can be used to drive specific, highly reductive biotransformations within industrially relevant hosts.

## Discussion

Here we show that the Mtr pathway can specifically deliver electrons to intracellular oxidoreductases and can drive intracellular redox reactions in a stoichiometric manner. Upon fumarate addition, *E. coli* take up electrons from the cathode via the MtrCAB complex and pass them to the fumarate reductases (Fig 6). The amount of current consumed can be increased by eliminating the MK reductase, Complex I (Fig 4). Interestingly, the current consumed is not stoichiometrically related to the accumulation of succinate from fumarate, but is

stoichiometric with reduction of nitrate to nitrite (Fig 5). Taken together, this work demonstrates use of *cymAmtrCAB* as a genetic module to stoichiometrically drive specific intracellular redox reactions in heterologous hosts. Below, we discuss the implications of this work for designing genetic modules for coupling cathode oxidation to intracellular reductions and opportunities for modulating cell behavior.

The finding that the Mtr pathway can be used to drive reductions by inner membrane oxidoreductases provides new opportunities to power key biological processes with electricity in a variety of microorganisms. For example, since the chemolithoautotroph *Nitrosomonas europaea* can use ammonia as its sole energy source and reductant [51], the production of ammonia via electricity, Mtr, and nitrate/nitrite reductases could be used to produce *N. europaea* biomass. Alternatively, we envision that, under microaerobic conditions, the Mtr pathway could be used to drive intracellular reduction of $O_2$ to water by cytochrome *bd* [52], which would generate a proton motive force and in turn make ATP. In these approaches, as well as others, the modularity and molecular-level specificity of the Mtr pathway allows rational design of strategies to precisely target electronic control of intracellular processes with minimal off-target effects–a long-sought goal of bioelectronics.

Our work elucidates several key points on how the MtrCAB module guides electrons out of and into *Escherichia coli* (Fig 6) but leaves additional points to be clarified. We demonstrate that the MtrCAB complex is needed and that electrons only reach the intracellular electron acceptor via specific oxidoreductase (Figs 2 and 3). However, it is unclear how electrons transverse the periplasmic space. In these strains, MtrA is present in the periplasm, making it plausible that MtrA shuttles between MtrC and oxidoreductases. Alternatively, a native *E. coli* protein may be serving as electron carrier to a *S. oneidensis* cyt *c*. Testing these possibilities will be the subject of future work.

## Conclusions

We demonstrate here that the *mtrCAB* genetic module delivers electrons from a cathode to specific oxidoreductases so that reductions can be driven stoichiometrically in a non-native host. This finding opens new opportunities to modulate key biological processes with electrodes using a strategy that can be extended to many microorganisms.

## Supporting information

**S1 Methods.**
(DOCX)

**S1 Table. Strains used in this study.**
(PDF)

**S2 Table. Plasmids used in this study.**
(PDF)

**S3 Table. Primers used in this study.**
(PDF)

**S4 Table. gBlocks used in this study.**
(PDF)

**S1 Raw images. Raw images of all gels and blots presented in this study.**
(PDF)

**S1 Fig. Electrochemical signatures of biotic fumarate reduction.** (A) Chronoamperometry upon fumarate addition to abiotic of bioelectrochemical reactors (black) and reactors containing CymAMtr-*E. coli* (orange), showing no sustained change in current in abiotic reactors. (B-D) Cyclic voltammetry (CV) measurement of bioelectrochemical reactors (B) without *E. coli*, (C) with the Ccm-*E. coli* strain, and (D) with the CymAMtr-*E.coli*. Black and orange lines represent the CV before and after 50 mM fumarate addition, respectively. Only reactors containing the CymAMtr-*E. coli* show a significant catalytic wave, which is located at -350 mV$_{Ag/AgCl}$.
(PDF)

**S2 Fig. Heterologous co-expression of CymAMtr and FrdABCD in *ΔfrdΔsdh* mutant.** (A) Images of *E. coli* cell pellets from the CymAMtr-*E.coli*, CymAMtr-*Δfrd Δsdh* and CymAMtr *frd$^+$Δsdh* after aerobic growth in 2xYT in the presence of IPTG. Expression of FrdABCD in the CymAMtr-*Δfrd Δsdh* mutant results in diminished red color of the bacteria, indicating a low abundance of matured cyts *c*. (B) Enhanced chemiluminescence (ECL) analysis of cyts *c* in the CymAMtr-*ΔfrdΔsdh*, CymAMtr-*frd$^+$Δsdh*, CymAMtr$^s$-*frd$^+$Δsdh* after aerobic growth in 2xYT in the presence of IPTG. These data indicate that introduction of a third plasmid to complement *frd* abrogates expression of the Mtr cyt *c*. However, regulating transcription of *cymAmtr-CAB* by the dynamic promoter ecpD (Boyarskiy et al., 2016) restores Mtr cyt *c* expression. (C) ECL analysis of cyts *c* in the Mtr$^s$-*frd$^+$Δsdh* and Mtr$^s$-*ΔfrdΔsdh* just before inoculation into the bioelectrochemical reactors and 7 days after fumarate was added to the reactors. As a control the cyts *c* expression was examined Mtr$^s$-*E. coli* and Ccm-*E.coli* (the two left lanes) which were grown in the same condition as the tested strain pre inoculation. (D) Growth curves of strains grown anaerobically in minimal medium supplemented with non fermentable glycerol, as the electron donor, and fumarate as the electron acceptor.
(PDF)

**S3 Fig. The influence of starvation on current production and the effect of Complex I disruption on cyt *c* expression.** (A) Chronoamperometry of CymAMtrCAB-*E. coli* upon addition of fumarate after 1 day (black), 3 days (orange), and 7 days (blue) of carbon-source deprivation, showing that increasing starvation also increases current consumption. (B) ECL analysis of the CymA, MtrC, and MtrA abundance in the CymAMtr-*E. coli* and CymAMtr-*ΔnuoH* strains when inoculated into the bioelectrochemical reactor (0 days) and 3 days after after addition of fumarate (3 days). As a negative control, the expression level of Ccm-*E. Coli* and Ccm-*ΔnuoH* strain is shown. Those strains were grown in the same condition as the tested strain pre-inoculation.
(PDF)

**S4 Fig. MenC is essential for fumarate respiration under anaerobic condition in CymAMtr-*E. coli* strain.** (A-B) Growth curves of strains grown in an anaerobic in minimal medium containing glycerol as the electron donor. No electron acceptor is provided (A) or Fumarate is provided as an electron acceptor (B). (C) Agarose gel electrophoresis of the PCR products obtained from the genomic DNA from CymAMtr-*E. coli*, CymAMtr-*ΔmenC* and CymAMtr-*menC$^S$* strains.
(PDF)

**S5 Fig.** (A) Cyclic voltammetry measurement showing no sustained change in current between abiotic reactors that don't contain or contain 40mM pyruvate (B) Representative HPLC chromatogram of a sample from the supernatant of a bioreactor after 14 days of CymAMtr-*E. coli* incubation. The chromatogram displays the peaks of the following analytes: pyruvate, malate, succinate, formate, acetate and fumarate. (C) A plot representing the supernatant

acetate, pyruvate malate and fumarate concentration in Bias (Black) vs unbiased (orange) bioreactors containing the Mtr-*ΔnuoH* mutant. Measurement started upon 50mM Fumarate addition and were taken over a period of 14 days values of 0.0 on plots indicating that the concentration was below the detection limit in that sample. All values are means of triplicate bioreactors, and error bars represent standard error. (D) A plot representing the supernatant succinate, formate, acetate, pyruvate malate and fumarate concentration in Bias(Black) vs unbiased (orange) bioreactors containing the Mtr-*E. coli* strain. Measurement started upon 50mM Fumarate addition and were taken over a period of 14 days values of 0.0 on plots indicating that the concentration was below the detection limit in that sample. All values are means of triplicate bioreactors, and error bars represent standard error.
(PDF)

**S6 Fig. Chronoamperometry of bioelectrochemical reactors containing CymA-Mtr E. coli (black) and CymAMtr-Δ*menC* (orange) upon nitrate addition, showing no significant change in the current consumed.** Red arrow indicates addition of nitrate to 10 mM, and the error bars indicate the standard deviation in current from triplicate bioelectrochemical reactors.
(PDF)

## Acknowledgments

We thank Prof. Michaela TerAvest, Prof. Jeff Gralnick and Dr. Joshua Atkinson for helpful conversations and Prof. Danielle Tullman-Ercek and Kersh Thevasundaram for help with the stress-responsive promoters.

## Author Contributions

**Conceptualization:** Moshe Baruch, Caroline M. Ajo-Franklin.

**Data curation:** Moshe Baruch, Sara Tejedor-Sanz, Lin Su.

**Formal analysis:** Moshe Baruch, Lin Su, Caroline M. Ajo-Franklin.

**Investigation:** Moshe Baruch, Sara Tejedor-Sanz, Lin Su.

**Methodology:** Moshe Baruch.

**Resources:** Caroline M. Ajo-Franklin.

**Visualization:** Moshe Baruch.

**Writing – original draft:** Moshe Baruch, Caroline M. Ajo-Franklin.

**Writing – review & editing:** Moshe Baruch, Sara Tejedor-Sanz, Lin Su, Caroline M. Ajo-Franklin.

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
