## [Decision Letter · Decision Letter 0]

1 Jul 2021

PONE-D-21-17413

Precise electronic control of redox reactions inside Escherichia coli using a genetic module

PLOS ONE

Dear Dr. Ajo-Franklin,

Thank you for submitting your manuscript to PLOS ONE. After careful consideration, we feel that it has merit but does not fully meet PLOS ONE’s publication criteria as it currently stands. Therefore, we invite you to submit a revised version of the manuscript that addresses the points raised during the review process.

In preparing your revised manuscript, please consider/address all questions and concerns raised by both reviewers.

We look forward to receiving your revised manuscript.

Kind regards,

Patrick C. Cirino

Academic Editor

PLOS ONE

Additional Editor Comments:

In preparing your revised manuscript, please consider/address all questions and concerns raised by both reviewers.

Journal Requirements:

3. Please amend the manuscript submission data (via Edit Submission) to include author Lin Su.

5. Please include captions for ALL your Supporting Information files at the end of your manuscript, and update any in-text citations to match accordingly. Please see our Supporting Information guidelines for more information: http://journals.plos.org/plosone/s/supporting-information.

Reviewers' comments:

Reviewer's Responses to Questions

**Comments to the Author**

1. Is the manuscript technically sound, and do the data support the conclusions?

Reviewer #1: Yes

Reviewer #2: Yes

2. Has the statistical analysis been performed appropriately and rigorously? 

Reviewer #1: Yes

Reviewer #2: I Don't Know

3. Have the authors made all data underlying the findings in their manuscript fully available?

Reviewer #1: Yes

Reviewer #2: Yes

4. Is the manuscript presented in an intelligible fashion and written in standard English?

Reviewer #1: Yes

Reviewer #2: Yes

5. Review Comments to the Author

Reviewer #1: In this work, the authors clearly demonstrate that E. coli modified to express electron uptake pathways from S. oneidensis are functional for reduction of specific electron acceptors. In addition, the work shows that unknown electron carriers and unknown electron sinks within the cell change the stoichiometry of redox reactions and likely affect the redox state of the cell. These are important steps towards understanding how to engineer extracellular electron uptake for biotechnology applications. I found the paper to be well-written and straightforward with the exception of the section describing results with the CymAMtr-ΔnuoH strain. Below are some questions for clarification on this section as well as some minor points.

The results with the CymAMtr-ΔnuoH strain are striking, but also a bit confusing to comprehend. I have a number of questions below for which I would be grateful if the authors would clarify in the text.

1. Line 270 - The number of c cyts was much lower in the CymAMtr-ΔnuoH strain. Is there a growth defect for this strain? Some reason why disrupting Complex I would affect all c cyt? On line 284 the authors note that they must provide pyruvate to sustain viability in the bioelectrochemical experiments but was pyruvate always added to this strain? I may just be naïve about E.coli growth metabolism but if you take out Complex I don’t you have to grow fermentatively?

2. Line 273-274 – This statement is a bit confusing. Are the authors trying to say that electrons from reduced NADH normally enter Complex I and provide reducing equivalents for fumarate reduction during electron uptake from the cathode? Where are the electrons coming from? Pyruvate fermentation?

3. Line 282 – Was the CymAMtr-ΔnuoH strain used for subsequent fumarate/succinate stoichiometry because it had higher electron uptake?

4. Line 490 – I think I am missing something…what does it mean that succinate consumption is “equal under polarized and unpolarized conditions”? Succinate is coming from fumarate so is it also the assumption that fumarate is reduced equally? I think this gets back to my question 3 above. Is there an electron donor in the system other than the electrode? Are the authors assuming pyruvate?

Minor points

Line 58 – do authors mean -560 mV?

Line 114 – Can authors clarify the difference between a technical and biological replicate? The figures show “replicates” but it is not clear what this means based on this statement in the methods.

Line 132 – missing word?

Line 139 – references Fig 1A but that is not what is depicted.

Line 146 – It isn’t really clear why strains need to be grown anodically before testing cathodically. Is this just to validate EET?

Line 157 – Are you sure there are no changes in gene expression? What about the fumarate reductase?

Line 163 – C43(DE3) is not really described before this point and the reader is left to assume that this is the strain used here.

Reviewer #2: Baruch et al. investigate electron uptake in E. coli engineered to express Mtr pathwawy components from S. oneidensis. Electron uptake to fumarate and nitrate could be demonstrated, the latter case seems to be stoichiometric conversion to ammonia (which is nice!). One of the major challenges in this system is the very low levels of current produced / consumed and the very large number of planktonic cells (0.6 OD). Overall, the findings are interesting and the work is sound. A better job can be done with the background on the known EET pathway of S. oneidensis. I would also like to see a little more detail in the main text regarding the claim of stoichiometric conversion of nitrate to ammonia. Comments for the authors consideration are below.

Line 1 – the authors may wish to provide a more accurate title. I don’t see how this control is ‘precise’ in nature.

Line 27 –Fumarate is intracellular, but nitrate is periplasmic.

Line 53 – the conversion of lactate to pyruvate will send electrons directly into the quinone pool or will produce NADH (depending if D-lactate or L-lactate is being metabolized). Conversion of pyruvate to acetate generates formate anaerobically, not NADH.

Line 56 – both CctA (also known as Stc) and FccA move electrons between CymA and MtrCAB. See reference 22 here. Mutants lacking cctA have no discernable metal or electrode reduction phenotype.

Line 59 – the references here are confusing. One reference shows electron uptake to oxygen and the other to fumarate under anoxic conditions?

Figure 1B – FccA does not directly interact with the menaquinone pool. It received electrons from CymA.

Line 76 – Unclear why differences in OM permeability (the references suggests, but does not demonstrate, that B strains have larger OM porins) would influence electron transfer across the outer membrane. The authors have not described why type II secretion is important to electron transfer.

Line 83 – why not include all methods here for completeness? Is this a space constraint from a past submission?

Line 146 – unclear why anaerobic, anodic incubation is required to prepare E. coli for cathodic conditions.

Line 162 – is it appropriate to cite an un-reviewed biorxiv paper?

Line 163 – the reference cited here does not provide information related to NapC complementing CymA.

Line 186 – was a CymAMtr strain with sdh missing also tested (I don’t’ think it need to be, but I’m curious)? Did these new mutant strains exhibit any phenotype on the anode, assuming they were pre-grown in this fashion as previously done?

Line 207 – unclear why the authors are concerned about polarity given that these are deletion mutants? Could this figure be moved to supplemental to help focus the work?

Line 234 – you aren’t comparing current production to wild-type – be specific!

Line 239 – CymA in S. oneidensis requires a MK co-factor to function and is also required to reduce FccA, the periplasmic fumarate reductase. It doesn’t seem surprising to me that the menaquinone mutants still exhibited cathodic fumarate reduction. There is likely reduced ubiquinone that is facilitating reduction of the reductase complexes.

Line 240 – unless there is free MtrA in the periplasm for some reason, considering the structure of the MtrCAB complex, it should not be able to get anywhere near cytoplasmic membrane complexes like Fdh and Sdh.

Line 271 – specify what ‘Mtr cyt c’ is being referred to here.

Line 370 – is E. coli a ‘novel microorganism’?

Line 378 (and 416) – the authors have presented two cases for driving reductive reactions. One case was driven stoichiometrically and the other was not. For the nitrate to ammonia example, the authors need to better walk through the math to convince the reader that it is indeed stoichiometric. How much ammonia was generated? Seems like ~ 0.35 mM. How many electrons would need to be consumed to produce this? How much current was in fact consumed over this time?

Line 393 – the heading here seems unnecessary

Line 410 – what is the evidence that supports the statement here, that MtrA is more abundant than MtrC in this system?

Figure 6 – is this figure useful? Also, I’m pretty sure the nitrate / nitrite reactions occur in the periplasm, not the cytoplasm.

Table S1 – please add the parent strain (and its complete genotype) to this list.

References – missing italics throughout, some are incomplete.

6. PLOS authors have the option to publish the peer review history of their article (what does this mean?). If published, this will include your full peer review and any attached files.

Reviewer #1: **Yes: **Sarah Glaven

Reviewer #2: No

---

## [Author Response · Author response to Decision Letter 0]

24 Aug 2021

We uploaded the Response to the Reviewers as a separate file as directed in the Decision Letter.

There are no gels or blots shown in the main Figures. The Supporting Information contains several images of gels and blots. These images are the full uncropped and unadjusted images.

---

## [Decision Letter · Decision Letter 1]

27 Sep 2021

Electronic control of redox reactions inside Escherichia coli using a genetic module

PONE-D-21-17413R1

Dear Dr. Ajo-Franklin,

We’re pleased to inform you that your manuscript has been judged scientifically suitable for publication and will be formally accepted for publication once it meets all outstanding technical requirements.

Kind regards,

Patrick C. Cirino

Academic Editor

PLOS ONE

Additional Editor Comments (optional):

Reviewers' comments:

Reviewer's Responses to Questions

**Comments to the Author**

1. If the authors have adequately addressed your comments raised in a previous round of review and you feel that this manuscript is now acceptable for publication, you may indicate that here to bypass the “Comments to the Author” section, enter your conflict of interest statement in the “Confidential to Editor” section, and submit your "Accept" recommendation.

Reviewer #1: All comments have been addressed

Reviewer #2: All comments have been addressed

2. Is the manuscript technically sound, and do the data support the conclusions?

Reviewer #1: Yes

Reviewer #2: Yes

3. Has the statistical analysis been performed appropriately and rigorously? 

Reviewer #1: Yes

Reviewer #2: N/A

4. Have the authors made all data underlying the findings in their manuscript fully available?

Reviewer #1: Yes

Reviewer #2: (No Response)

5. Is the manuscript presented in an intelligible fashion and written in standard English?

Reviewer #1: Yes

Reviewer #2: Yes

6. Review Comments to the Author

Reviewer #1: (No Response)

Reviewer #2: (No Response)

7. PLOS authors have the option to publish the peer review history of their article (what does this mean?). If published, this will include your full peer review and any attached files.

Reviewer #1: **Yes: **Sarah Glaven

Reviewer #2: No

---

## [Editor Report · Acceptance letter]

20 Oct 2021

PONE-D-21-17413R1 

Electronic control of redox reactions inside *Escherichia* coli using a genetic module 

Dear Dr. Ajo-Franklin:

I'm pleased to inform you that your manuscript has been deemed suitable for publication in PLOS ONE. Congratulations! Your manuscript is now with our production department. 

Kind regards, 

on behalf of

Dr. Patrick C. Cirino 

Academic Editor

PLOS ONE